# Evaluation of NS4A, NS4B, NS5 and 3′UTR Genetic Determinants of WNV Lineage 1 Virulence in Birds and Mammals

**DOI:** 10.3390/v15051094

**Published:** 2023-04-29

**Authors:** Lise Fiacre, Steeve Lowenski, Céline Bahuon, Marine Dumarest, Bénédicte Lambrecht, Maha Dridi, Emmanuel Albina, Jennifer Richardson, Stéphan Zientara, Miguel-Ángel Jiménez-Clavero, Nathalie Pardigon, Gaëlle Gonzalez, Sylvie Lecollinet

**Affiliations:** 1Animal Health Laboratory, L’alimentation et L’environnement (INRAE), Institut National de Recherche pour L’agriculture, École Vétérinaire d’Alfort (ENVA), Agence Nationale de Sécurité Sanitaire de L’alimentation, de L’environnement et du Travail (ANSES), UMR Virology, 94700 Maisons-Alfort, France; 2Centre de Coopération Internationale en Recherche Agronomique pour le Développement (CIRAD), UMR ASTRE, 97170 Petit-Bourg, France; 3ASTRE, CIRAD, INRAe, University of Montpellier, 34000 Montpellier, France; 4SCIENSANO, Avian Virology and Immunology, 1180 Brussels, Belgium; 5Centro de Investigación en Sanidad Animal (CISA-INIA), CSIC, Carretera Algete-El Casar s/n, 28130 Valdeolmos, Spain; 6CIBER Epidemiología y Salud Pública (CIBERESP), 28001 Madrid, Spain; 7Institut Pasteur, URE ERI/CIBU, 75015 Paris, France

**Keywords:** West Nile virus, molecular determinants, virulence, reverse genetics

## Abstract

West Nile virus (WNV) is amplified in an enzootic cycle involving birds as amplifying hosts. Because they do not develop high levels of viremia, humans and horses are considered to be dead-end hosts. Mosquitoes, especially from the *Culex* genus, are vectors responsible for transmission between hosts. Consequently, understanding WNV epidemiology and infection requires comparative and integrated analyses in bird, mammalian, and insect hosts. So far, markers of WNV virulence have mainly been determined in mammalian model organisms (essentially mice), while data in avian models are still missing. WNV Israel 1998 (IS98) is a highly virulent strain that is closely genetically related to the strain introduced into North America in 1999, NY99 (genomic sequence homology > 99%). The latter probably entered the continent at New York City, generating the most impactful WNV outbreak ever documented in wild birds, horses, and humans. In contrast, the WNV Italy 2008 strain (IT08) induced only limited mortality in birds and mammals in Europe during the summer of 2008. To test whether genetic polymorphism between IS98 and IT08 could account for differences in disease spread and burden, we generated chimeric viruses between IS98 and IT08, focusing on the 3′ end of the genome (NS4A, NS4B, NS5, and 3′UTR regions) where most of the non-synonymous mutations were detected. In vitro and in vivo comparative analyses of parental and chimeric viruses demonstrated a role for NS4A/NS4B/5′NS5 in the decreased virulence of IT08 in SPF chickens, possibly due to the NS4B-E249D mutation. Additionally, significant differences between the highly virulent strain IS98 and the other three viruses were observed in mice, implying the existence of additional molecular determinants of virulence in mammals, such as the amino acid changes NS5-V258A, NS5-N280K, NS5-A372V, and NS5-R422K. As previously shown, our work also suggests that genetic determinants of WNV virulence can be host-dependent.

## 1. Introduction

West Nile virus (WNV) is a flavivirus belonging to the family *Flaviviridae*, like Dengue virus (DENV) and Zika virus (ZIKV), and belongs to the Japanese encephalitis virus (JEV) serocomplex. It was first isolated in 1937 from the blood of a woman at Omogo, in the West Nile district of North-Western Uganda, in Africa [1]. The WNV genome is a positive sense single-stranded RNA (+ssRNA) of around 11 kb long, encoding three structural proteins (C, prM, E) and seven non-structural (NS) proteins (NS1, NS2A, NS2B, NS3, NS4A, NS4B, NS5) generated from a single polyprotein [2]. WNV is maintained in a mosquito-bird-mosquito enzootic transmission cycle, with occasional spillovers to accidental dead-end hosts, humans and horses [3]. Most human infections are asymptomatic, but in the case of symptomatic infections, patients generally suffer from mild febrile illness associated with headaches or fatigue [4]. In severe forms of WNV infection, however, neuroinvasion leads to encephalitis, meningitis, or acute flaccid paralysis [5]. Elderly and immunocompromised persons are at increased risk for development of neuroinvasive West Nile disease [2]. While most human cases occur after mosquito bites, direct human–human transmission has been described after blood transfusion [6], solid organ transplantation [7], or transplacental exposure (CDC). The first case of West Nile neuroinvasive disease acquired after liver transplantation was reported in 2002 in Michigan [8]. In the natural setting, WNV is transmitted by mosquito vectors belonging to the genus *Culex (Cx.)*, such as *Cx. univittatus* in Africa or the Middle East, *Cx. pipiens, tarsalis*, and *quinquefasciatus* in North America, and *Cx. pipiens* and *Cx. modestus* in Europe [9]. The vector competence of *Aedes (Ae.) albopictus*, belonging to a different mosquito genus, has only been demonstrated experimentally [10]. During its transmission cycle, WNV is able to infect a large variety of bird species, but only avian species belonging to Passeriformes, Falconiformes, Strigiformes, and Charadriiformes appear to be competent vertebrate hosts, with viremia levels high enough to support transmission of WNV to mosquitoes [11].

Before 1996, WNV silent or paucisymptomatic infections had been reported in Africa, Europe, and the Middle East. WNV lineage 1 strains (re)emerged in the 1990s in Romania (1996), Israel (1997–2001), Italy (1998), Russia (1999), and since 1999 in the United States (US), causing more cases of neurological disease in humans and in horses [12,13]. At the end of the summer of 1999, clustered fatalities were reported in American crows (*Corvus brachyrhynchos*), fish crows (*Corvus ossifragus*), and captive birds in the Bronx Zoo, in New York [14]. Meanwhile, an increasing frequency of encephalitis was observed in humans. The two events were finally attributed to WNV transmission and neuropathology, as a result of virus detection and sequencing [14,15,16]. Since 1999, WNV has been responsible for 56,000 reported human cases including more than 28,000 neuroinvasive cases and 2700 deaths, underscoring WNV as one of the most important zoonotic diseases in North America [17]. Phylogenetic analysis showed that the US NY99 strain was most closely related (> 99.8% nucleotide identity) to a virus strain isolated from the brain of a dead goose in Israel in 1998 (IS98) [15], suggesting that WNV emergence in the US arose from viral introduction from the Middle East (whether through anthropogenic activities supporting the introduction of infected vectors or birds, or through bird migration).

WNV has been (re)emerging throughout Europe over the last 15 years, associated with the spread of recently-introduced lineage 2 strains. The year 2018 was marked by an exceptional upsurge of WNV outbreaks in Europe, with 2083 autochthonous human cases and 187 deaths reported in 15 different countries [18]. In 2022, Europe experienced another intense WNV transmission season: 1191 human cases of WNV infection were identified in 11 countries, including 586 in northern Italy, where newly introduced lineage 1 virus and lineage 2 strains co-circulated [19]. These observations highlight the need for better characterization of determinants of WNV virulence and pathogenesis. Future WNV outbreaks are particularly difficult to predict, as they depend on multiple environmental factors including temperature, humidity, mosquito distribution, abundance and diversity, resident bird populations, and bird migration routes [20,21,22], as well as on intrinsic viral factors (strain virulence, replication properties, etc.). Novel methodologies are needed to enhance preparedness and control, should a highly virulent strain of WNV emerge in Europe. By identifying virulence markers of European and Mediterranean WNV strains, new screening tools could be developed to predict the virulence of emerging strains and anticipate their spread, and safe and effective attenuated WNV vaccines for humans could be developed [23]. Six human vaccine candidates have entered phase 1 and 2 clinical trials, but none are commercially available [24,25,26,27,28,29,30].

Researchers have long been using molecular technologies including reverse genetics for precise modification of the virus genome and creation of WNV mutants or chimeras to identify residues involved in WNV virulence; most of these studies have been carried out with the NY99 strain [31,32], while additional data for European Mediterranean and African lineage 1 and 2 virulent strains are strongly needed [33].

Israel experienced intense and recurrent WNV epidemics in 1997–2001 (35 and 417 human neuroinvasive cases in 1998 and 2000, respectively), associated with the isolation of a highly virulent lineage 1 strain belonging to the Israelo-American cluster [34,35]. Of note, the Israel 1998 (IS98) WNV strain displays more than 99% identity with the NY99 strain and has been found to be highly virulent in domestic and wild birds, leading to high fatality rates in geese and white storks. In 2008, a new epidemic lineage 1 strain emerged in Italy (IT08), causing little mortality in wild birds. These observations suggest that the two Mediterranean lineage 1 strains IS98 and IT08 have genetic differences driving differential virulence for birds. Dridi et al. [36] tested this hypothesis by infecting specific-pathogen free (SPF) chicks either with IS98 strain or IT08 strain and showed a significant increase in the mortality of chicks infected with IS98 through the intracranial route.

WNV non-structural (NS) proteins play key roles in virus replication and the control of host immune responses, as well as in virus virulence [31]. In this regard, the NS4A protein is involved in viral replication and regulates the helicase activity of NS3 [37,38]. NS4B is specifically involved in immune evasion by acting on the type I interferon (IFN) signaling pathway [39]. NS4A and NS4B proteins have been shown to influence WNV virulence in mammal hosts [40,41], avian hosts [42], and in mosquito vectors [43], and have also been identified as virulence factors in other flaviviruses, including DENV [44], JEV [45], or yellow fever virus (YFV) [46]. The NS5 WNV protein is the virus polymerase and methyltransferase, which is critical in viral replication and is recognized as a virulence determinant for WNV in mammalian hosts [47]. Few studies have addressed the importance of the 3′-untranslated region (UTR) region in WNV virulence [48]. In this context, the present study aimed to gain better understanding of the roles of NS4, NS5, and 3′UTR regions in WNV virulence, using chimeric constructs between NS4/NS5/3′UTR regions of high- and low-virulence strains for birds, IS98 and IT08, respectively, and more particularly, to identify which regions were responsible for the difference in virulence observed between these two strains [36].

## 2. Materials and Methods

### 2.1. Cell Lines and Viruses

Viral production, as well as the study of growth kinetics, was achieved by using Vero cells. Growth kinetics were also assessed on CCL141 cells. Vero cells (ATCC CCL-81) were maintained at 37 °C, 5% CO_2_ in Dulbecco modified Eagle’s medium (DMEM, Thermo Fisher Scientific) supplemented with 5% fetal bovine serum (FBS, Lonza), 1 mM sodium pyruvate, and penicillin (1 U/mL)/streptomycin (1 µg/mL) (Thermo Fisher Scientific). The CCL141 cell line (provided by Nolwenn Jouvenet, Institut Pasteur de Paris) was maintained at 37 °C, 5% CO_2_ in DMEM supplemented with 10% FBS, Lonza, and antibiotics as described above.

The infectious cDNA clone derived from IS98, a highly virulent strain isolated from a white stork, *Ciconia ciconia* (IC-WNV-IS98; Genbank accession number: KR107956.1), was constructed as described by Bahuon et al., and cleared virus supernatants obtained after a single passage in Vero cells were used in in vitro and in vivo infections [49]. The parental WT-WNV-Italy 2008 15803 strain, WNV-IT08 (Genbank accession number: FJ483549.1), was isolated from a magpie, *Pica pica*, in 2008 at IZSLER and passaged once in Vero cells. This first-passage virus was used in the following experiments.

### 2.2. Generation of Chimeric Viruses

The chimeric viruses were generated using protocols adapted from Bahuon et al. [49]. Briefly, genomic RNA from WNV-IT08 was extracted using the QIAmp viral RNA kit (Qiagen, Hilden, Germany). Two specific RNA fragments covering 2 regions of the 3′-end of the virus genome were reverse transcribed using the OneStep RT-PCR Kit (Qiagen, Hilden, Germany). Two genomic fragments were amplified using primers 5761F 5′-CGTGCTGGAAAGAAAGTAGTC-3′, 8100R 5′-GTAGAACACATCCACTCCACTC-3′ for the first chimera and 7993F 5′-GAAGTCAGAGGGTACACAAAGG-3′, 10894R 5′-TCCTTCCCCTGACCTACA-3′ for the second chimera (positions can be found on the virus genome in Figure 1). These were individually cloned into the pCR2.1 plasmid and then subcloned into the IC-WNV-IS98 plasmid following enzymatic digestion to generate cDNA clones of two chimeric viral genomes, IS98-NS4A/NS4B/5′NS5 and IS98-NS5/3′UTR. These were propagated in *E. coli* DH5α (Thermo Fisher Scientific) at room temperature. Corresponding plasmids were linearized and transcribed in vitro using the Sp6 MEGAscript High yield transcription kit (Ambion, Thermo Fisher Scientific, Montigny-le-Bretonneux, France). Vero cells were electroporated with 20 µg of RNA transcripts, before being mixed with DMEM supplemented with 2% FBS and incubated in a T-25 flask (5% CO_2_ at 37 °C) until observation of a cytopathic effect (CPE). This is referred to as a passage 0, P0. At day 2 post-infection (pi), 1 mL of P0 supernatants was used to infect new Vero cells in T-25 flasks (referred to as passage 1, P1). Three days pi, after observation of CPE, viruses were harvested, and supernatants were aliquoted and stored at −80 °C. The 2 chimeric viruses were titrated by plaque assay on Vero cells.

### 2.3. WNV Plaque Phenotype

Vero cells were seeded in 6-well plates and infected with either the IC-WNV-IS98, parental WT-WNV-IT08 strain or chimeric viruses IS98-NS4A/NS4B/5′NS5 and IS98-NS5/3′UTR at 20 PFU for 1 h 30 min at 37 °C, 5% CO_2_. The inoculum was discarded, and cells were overlaid with 2% SeaPlaque agarose in MEM 1×, supplemented with 5% FBS, 1% sodium pyruvate, 1 U/mL of penicillin and 1 mg/mL of streptomycin, and incubated for 3 days at 37 °C, 5% CO_2_. Once the agarose was removed, cells were fixed with 4% paraformaldehyde and stained with 0.4% crystal violet for 24 h in a humid chamber at 37 °C.

### 2.4. Viral Growth Kinetics

Vero and CCL141 cells were seeded in 12-well plates and infected with IC-WNV-IS98, parental WT-WNV-IT08 strain or chimeric viruses IS98-NS4A/NS4B/5′NS5 and IS98-NS5/3′UTR at different MOIs (1, 0.1, and 0.01) in DMEM supplemented with 2% FBS. The cells were incubated at 37 °C, 5% CO_2_, and cell supernatants were harvested at the indicated times post-infection (17 h, 24 h, 48 h, and 72 h). Viral titers were determined by quantitative RT-PCR and TCID50 on Vero cells.

### 2.5. Quantitative RT-PCR

Total RNA was extracted using the MagVet Universal Isolation kit (Thermo Fisher Scientific, Montigny-le-Bretonneux, France) according to the manufacturer’s instructions. RNA was reverse-transcribed and amplified using the AgPath-ID One-Step RT-PCR Kit (Applied Biosystems, Thermo Fisher Scientific, Montigny-le-Bretonneux, France) as previously described [1]. For specific amplification of the viral genome, primers WNproC-10 5′-CCTGTGTGAGCTGACAAACTTAGT-3′ and WNproC-132 5′-GCGTTTTAGCATATTGACAGCC-3′ targeting the 5′UTR and C regions (total length: 123 nucleotides) were used with the probe 5′-FAM-CCTGGTTTCTTAGACATCGAGATCT-Tamra-3′ [50]. The primers ACTB-966 5′-CAGCACAATGAAGATCAAGATCATC-3′ and ACTB-1096 5′-CGGACTCATCGTACTCCTGCTT-3′ were used to amplify cellular RNA with the probe 5′-VIC-TCGCTGTCCACCTTCCAGCAGATGT-TAMRA-3′ (total length of the ACTB amplicon: 131 nucleotides) [51]. Primers and probes were used at a concentration of 0.4 µM and 0.2 µM, respectively. To perform the reaction, 5 µL of RNA was completed to a final volume of 25 µL with reaction mix. Amplification was performed in an AB 7300 Real-Time PCR system (Applied Biosystems, Thermo Fisher Scientific, Montigny-le-Bretonneux, France). The samples were maintained at 45 °C for 10 min and at 95 °C for 10 min and then subjected to 40 cycles consisting of incubations at 95 °C for 15 s and at 60 °C for 60 s. WNV and β-actin RNA standards were used to construct standard curves and quantify viral and cellular RNAs.

### 2.6. Virus Titration by TCID50

Virus titration was performed in 96-well flat bottom plates. Vero cells were seeded one day prior to infection at a density of 2 × 10^4^ cells/well. Quadruplicate wells were inoculated with ten-fold serial dilutions of virus supernatants (10^−1^ to 10^−8^). Plates were incubated for 3 days at 37 °C, 5% CO_2_, and 50% tissue culture infection dose (TCID_50_) was determined based on the number of wells displaying CPE at each dilution using the methodology described by Reed and Muench [52].

### 2.7. Virulence in Mice

Six-week-old Balb/cByJ mice (Charles River Laboratories, L’Arbresle, France) were housed in an environmentally controlled room under biosafety level 3 conditions and were given food and water ad libitum. Animal experiments were approved by the joint Anses-UPEC-Alfort Veterinary School ethic committee (permit number: 15/02/11–13). Seventeen groups of 5 mice were inoculated intraperitoneally with various doses of virus (0.1, 1, 10, and 100 PFU of viruses: WNV-IS98, WNV-It08, IS98-NS4A/NS4B/5′NS5, or IS98-NS5/3′UTR) prepared in PBS or with PBS alone as a negative control. Infected mice were checked at least daily for two weeks and humanely euthanized when the end point was reached, e.g., when at least two of the following clinical signs were observed: weight loss greater than 15%, anorexia, ruffled hair, curved back, loss of balance, or paresis. The presence of viral RNA was confirmed in blood collected at 3 days pi (dpi) and in brain tissue (data not shown), recovered shortly after their death, by RT-qPCR. Total RNA was extracted using the MagVet Universal Isolation kit (Thermo Fisher Scientific) according to the manufacturer’s instructions, from 100 µL of blood or 100 mg of brain tissue. To assess the induction of WNV-antibody in surviving animals at 21 dpi, blood was collected, and serum was separated after centrifugation for 5 min at 2044 RCF. ID Screen West Nile competition ELISA kit (ID Vet, Evreux, France) was performed on mice sera according to the manufacturer’s instructions and demonstrated the induction of WNV antibodies in all surviving mice infected with doses ≥ 1 PFU (data not shown).

### 2.8. Virulence in Specific-Pathogen-Free (SPF) Chickens

Animal experiments on SPF chickens were carried out according to Dridi et al. [36]. SPF chickens, provided by Lohmann Valo (Cuxhaven, Germany), were kept in biosecurity level 3 isolators at CERVA-CODA, and animal experiments were conducted under the authorization and supervision of the Bioethics Committee CODA-IPS (agreement number: LA1230174). Five groups of 10 one-day-old SPF chickens were intracerebrally (50 µL inoculum/chicken) inoculated with 10^3^ TCID_50_ of virus supernatants or of PBS (control group). Clinical evaluation was performed daily for 14 days. On day 3 pi, 3 chickens/group were sacrificed to collect larger volumes of blood and evaluate viremia by RT-qPCR. The presence of virus in oral swabs and feathers was assessed by RT-qPCR at day 3 and at days 3, 7, and 14 pi, respectively. Two to three wing feather follicles per bird were sampled and stored in 600 μL of RNA later (RNA Stabilization Reagent, Qiagen Benelux B.V., The Netherlands) at −80 °C for further RNA extraction, as described in [36]; the method used for RNA extraction from feather follicles is also detailed in [36]. Total RNA from 100 µL of oral swabs or blood samples was extracted using the MagVet Universal Isolation kit (Thermo Fisher Scientific) on the Kingfisher extraction robot (Thermo Fisher Scientific, Lyon, France).

Mortalities were recorded and cadavers removed daily from the isolators, while surviving chickens were finally euthanized at day 14 pi. The brain of dead chickens was sampled for RT-qPCR analysis.

### 2.9. Statistical Analysis

Statistical analysis of in vitro and in vivo results was performed using the non-parametric Kruskal–Wallis or *t*-test mean analysis. All analyses were performed using GraphPad Prism version 7.

## 3. Results

### 3.1. Comparison of Genome Sequences of Parental WNV-IS98 and IT08 Strains

The complete genome sequence of the WNV-IS98 strain (GenBank accession number AF481864.1) exhibits 446 nucleotide (nt) differences (4.0%) compared with that of the WNV-IT08 strain (GenBank accession number FJ483549.1). Nucleotide substitutions are distributed throughout the genome (5′UTR: 0 substitutions (0.0%), C protein gene: 6 substitutions (1.6%), prM-M protein gene: 20 substitutions (4.0%), E protein gene: 58 substitutions (3.9%), NS1 protein gene: 40 substitutions (3.8%), NS2A protein gene: 34 substitutions (4.9%), NS2B protein gene: 20 substitutions (5.1%), NS3 protein gene: 85 substitutions (4.6%), NS4A protein gene: 19 substitutions (4.3%), NS4B protein gene: 35 substitutions (4.6%), NS5 protein gene: 104 substitutions (3.8%), 3′UTR: 25 substitutions (4%)) (Figure 2). Within the coding sequence, WNV-IT08 and IS98 differ by 22 non-synonymous mutations (C protein: 0/123 non-synonymous mutation, prM-M protein: 0/167 non-synonymous mutation, E protein: 2/497 non-synonymous mutations (0.4%), NS1: 2/352 non-synonymous mutations (0.6%), NS2A: 3/231 non-synonymous mutations (1.3%), NS2B: 3/131 non-synonymous mutations (2.3%), NS3: 3/619 non-synonymous mutations (4.8%), NS4A: 2/149 non-synonymous mutations (1.3%), NS4B: 1/256 non-synonymous mutation (0.4%), NS5: 6/905 non-synonymous mutations (0.7%)) (Figure 2). While few non-synonymous mutations are found along the genome, WNV-IS98 and IT08 are known to display differential virulence and lethality in wild birds or SPF chicks [36], suggesting that some of these polymorphisms represent genetic determinants of virulence.

### 3.2. Viral Replication in Vero Cells

The efficacy of viral replication and cell-to-cell spread, and hence the size of viral plaques, is correlated with virulence. In particular, small plaques generally denote an attenuated phenotype, whereas large plaques often correspond to highly virulent strains [53]. Analysis of plaque size in Vero cells revealed that the IS98-NS5/3′UTR chimera generated smaller plaques (0.5 mm) than parental WNV-IS98 (1 mm), WNV-IT08 (1.5 mm), and the IS98-NS4A/NS4B/5′NS5 chimera (1.5 mm) at 3 dpi (Figure 3). In agreement with the assertion that small plaques would be associated with lower replication or less effective exocytosis, our results suggest that the IS98-NS5/3′UTR chimera could be less virulent than parental viruses. This hypothesis was confirmed by in vivo analysis in mice, in that IS98-NS5/3′UTR induced less mortality than WNV-IS98 (Figure 6). Of note, WNV-IT08 demonstrated lower virulence than WNV-IS98 in an avian model of WNV infection (SPF chicks), but did not show reduced virulence in Vero cells.

### 3.3. Replication Kinetics of the Parental WNV-IS98 and IT08 and Chimeras thereof in Mammalian and Avian Cell Culture

Vero cells, derived from kidney epithelial cells of an African green monkey, were selected as a mammalian cellular model for evaluation of replication kinetics. Serial samples of cell supernatants were collected at 17, 24, 48, and 72 h post-infection (hpi) and used to quantify total viral RNA (viral genome copies/μL) by RT-qPCR and infectious viral titers, by end-point dilution in Vero cells. Total viral RNA was significantly lower for IS98-NS5/3′UTR than for parental WNV-IS98 and IT08 at 17, 48, and 72 hpi (Kruskal–Wallis; *p* < 0.05 for WNV-IT08 at 17 hpi; *p* < 0.01 for WNV-IS98 at 48 and 72 hpi) (Figure 4). Thus, in agreement with the results of the plaque assay (Figure 3), replication kinetics in Vero cells suggests that the IS98-NS5/3′UTR chimera had impaired replication or exocytosis in comparison with parental WNV-IS98 or IT08 strains and the IS98-NS4A/NS4B/5′NS5 chimera. Nevertheless, no differences in infectious viral titers in Vero cell supernatants were observed between parental and chimeric viruses.

Replication kinetics of parental WNV-IS98 and IT08 and chimeric viruses was also evaluated on the avian cell line, CCL141, derived from duck embryo cells and exhibiting fibroblast morphology. Total viral RNA (viral genome copies/μL) was significantly lower for the IS98-NS5/3′UTR chimera than for parental WNV-IS98 and IT08 strains at 48 and 72 hpi (*p* < 0.05), while higher replication levels were observed for the IS98-NS4A/NS4B/5′NS5 chimera at 17 hpi only (*p* < 0.05 with WNV-IS98) (Figure 4). These results suggest that replication for the IS98-NS5/3′UTR chimera was diminished at late time points (48 and 72 hpi), as observed in Vero cells. Nevertheless, while viral titers for the IS98-NS5/3′UTR chimera tended to be lower than those of the parental WNV-IS98 and IT08 strains after 24 h, the difference did not reach statistical significance.

### 3.4. Replication of Parental and Chimeric Viruses in BALB/cByJ Mice

While cellular models can be informative regarding the intrinsic replicative capacity of viruses, virulence, being a complex trait, can only truly be assessed in vivo. To this end, we performed analyses in mice, as they are highly susceptible to WNV infection and develop clinical signs closely similar to those in humans, including such neurological features as paralysis. In vivo analyses also permit analysis of viremia, which reflects viral replication at the inoculation point and subsequent dissemination via the hematogenous route. Peak viremia typically occurs at 2 to 3 dpi.

At 3 dpi, at the expected peak of viremia, viral load was analyzed by RT-qPCR for mice infected with WNV-IS98, IT08, and the two chimeric viruses. Viremia was observed to be similar at 3 dpi for IC-WNV-IS98, WNV-IT08, and the chimera IS98-NS4A/NS4B/5′NS5 (Figure 5). The IS98-NS5/3′UTR chimera, however, induced a significantly lower viremia than the other tested viruses, suggesting a reduced replicative capacity for this chimera in mice (*p* < 0.05).

### 3.5. Neuroinvasion and Neurovirulence of Parental Strains and Chimeras in BALB/cByJ Mice

Neuroinvasion and neurovirulence properties of IC-WNV-IS98, WT-WNV-IT08, and chimeras IS98-NS4A/NS4B/5′NS5 and IS98-NS5/3′UTR were examined following intraperitoneal (i.p) inoculation of six-week-old female BALB/cByJ mice. Mice were monitored for 21 days.

At the dose of 1 PFU, all mice infected with IC-WNV-IS98, a highly virulent strain, died before 10 dpi. By contrast, some mice infected with WT-WNV-IT08 (60%) or by the chimeras IS98-NS4A/NS4B/5′NS5 (50%) and IS98-NS5/3′UTR (40%) survived the challenge, although these differences were not statistically significant. The median day of death was slightly different between the four groups: 8.7 for IC-WNV-IS98, 8.3 for WT-WNV-IT08, 8.2 for IS98-NS4A/NS4B/5′NS5, and 9.8 for IS98-NS5/3′UTR (Figure 6).

Moreover, infection with diminishing doses of virus (100, 10, 1, 0.1, and 0.01 PFU) allowed calculation of the lethal dose 50 (LD_50_) for the different viruses. The LD_50_ of IC-WNV-IS98 (0.3) tended to be lower than that of WT-WNV-IT08 (0.7) or of the two chimeras IS98-NS4A/NS4B/5′NS5 (1.0) and IS98-NS5/3′UTR (2.5), but such differences did not attain statistical significance.

Taken together, the survival curves and the LD_50_ determination tended to suggest that the IS98-NS5/3′UTR and IS98-NS4A/NS4B/5′NS5 chimeras were less virulent than WNV-IS98 in the mammalian model (Figure 6). In keeping with this conclusion, viral RNA load in the brain of WNV-infected mice was higher in the group inoculated with IC-WNV-IS98 (>10^5^ log_10_ copy RNA/µL) than in the other three groups (between 10^3^ and 10^4^ log_10_ copy RNA/µL) (data not shown).

In sum, the in vivo results in female BALB/cByJ mice show that both WNV-IS98 and IT08 parental strains, as well as chimeras thereof, are neuroinvasive and neurovirulent. In agreement with the lower viremia observed at day 3 after i.p. inoculation, the chimera IS-NS5/3′UTR tended to be less neurovirulent and neuroinvasive than the other evaluated viruses (Figure 6).

### 3.6. Replication of Parental and Chimeric Viruses in SPF Chickens

Birds are the principal amplifying hosts for WNV, and young chicks have been shown to be susceptible to WNV infection and to allow pathotyping of WNV strains. One-day-old SPF chickens were inoculated intracranially (i.c.) and subcutaneously (s.c) (data not shown), at a dose of 10^3^ TCID_50_ and monitored for 14 days pi. As previously shown, WNV strain virulence could be efficiently discriminated after i.c. inoculation in chicks [36]. After i.c. inoculation, peak viremia determined at 3 dpi was similar irrespective of which virus was considered (Figure 7a). Peripheral spread was estimated by RT-qPCR on crushed feather follicles obtained at 3 dpi. Viral RNA load was similar in groups infected with WNV-IS98 and the IS98-NS5/3′UTR chimera, while 1-log lower RNA quantities were reported from feathers sampled in chicks infected with WNV-IT08 and the IS98-NS4A/NS4B/5′NS5 chimera (not significant) (Figure 7b). At 3, 7, and 14 dpi, oral swabs were taken to estimate virus excretion levels, and viral RNA measured by RT-qPCR. Viral RNA was no longer detectable in oral swabs collected at 14 dpi (data not shown). While no significant differences were observed at 3 dpi, the parental IT08 strain and the IS98-NS4A/NS4B/5′NS5 chimera displayed significantly less viral RNA at 7 dpi than the WNV-IS98 parental strain (Figure 7c, *p* < 0.05).

### 3.7. Neurovirulence of Parental Strains and Chimeras in SPF Chickens

Neurovirulence of the parental strains and chimeras were compared by survival curve analyses (Figure 8a) and viral RNA quantification in brain by RT-qPCR (Figure 8b). When infected by the i.c. route, survival curves of WNV-infected SPF chicks showed significant differences between IC-WNV-IS98 and the IS98-NS4A/NS4B/5′NS5 chimera (*p* < 0.05). Although statistically significant differences were only observed between IC-WNV-IS98 and the chimera IS98-NS4A/NS4B/5′NS5, comparable survival curves were reported for WT-WNV-IS98 and the IS98-NS5/3′UTR chimera and for WT-WN-IT08 and the IS98-NS5/3′UTR (Figure 8a). Similar observations were obtained after viral RNA quantifications on feathers at 3 dpi and in oral swabs at 7 dpi (Figure 7c and Figure 8b).

Viral RNA quantification in the brain of dead chicks did not reveal any statistically significant difference among the studied viruses.

## 4. Discussion

The aim of this study was to gain understanding of the genetic determinants of WNV virulence, and in particular those that distinguish the highly virulent lineage 1 strain IS98 [49] and the less virulent lineage 1 strain IT08, isolated from wild birds during Mediterranean outbreaks of WNV that took place in 1998 and 2008, respectively. To this end, molecular chimeras of the WNV-IS98 and IT08 strains were generated by reverse genetics. The biological properties of parental and chimeric viruses were compared in mammalian and avian cell culture models, and virulence was assessed in vivo in both mammals and birds, and more particularly in mice and the 1-day old chick [36], respectively. We found that replacement of the IS98 sequence encoding NS4A, NS4B, and the 5′ portion of NS5 by corresponding sequences of the less virulent IT08 strain significantly diminished viral load in chicks at 7 dpi and improved survival, suggesting that certain nucleotide substitutions between the two parental viruses within the exchanged genomic region represent genetic determinants of WNV virulence in birds.

Comparison of WNV-IS98 and IT08 genomic sequences showed that around 40% of the 446 nucleotide substitutions that distinguish the viruses were located in the 3′ region of the genome, over a segment spanning NS4-3′UTR. Twenty-three percent of the 446 mutations were detected in the NS5 coding region. Only 22 of the 446 nucleotide substitutions represented non-synonymous mutations, and, of these, 9 were evidenced in the NS4-3′UTR segment, with 3 and 6 non-synonymous mutations in NS4A/B and NS5 coding sequences, respectively (Figure 2). Based on such findings, we hypothesized that NS4A/B and NS5 proteins were good candidates for investigation of the genetic determinants that distinguish the virulence of WNV-IS98 and IT08.

Our hypothesis was also supported by previous studies that had identified virulence markers in WNV NS proteins. Around two thirds of the genome encode NS proteins, which are essential for virus replication. NS4A and B regulate the ATPase activity of the NS3 helicase and play a role in viral replication [54] and in immune evasion through inhibition of the type-I IFN pathway [55]. NS5 is composed of an N-terminal methyltransferase (MTase) and a C-terminal RNA-dependent RNA polymerase (RdRp) [56] domain. Many mutations at the 3′ end of the genome, including NS4A-Q46K [37], NS4A-Q47K [37], NS4A-D50K [37], NS4B-P38G [41], NS4B-C102S [41], NS5-K61A [57], NS5-D146A [57], NS5-K182A [57], NS5-E218A [47], NS5-A804V [48], 3′UTR-A10596G [48], 3′UTR-C10774 [48], and 3′UTR-A10799G [48], have previously been shown to have an impact on WNV virulence in mammals. In birds, however, molecular determinants modulating WNV virulence have been poorly characterized. At the 3′ end of the genome, only a single mutation, NS4A-F92L, has previously been described as affecting avian virulence in birds [58]. The NS5 protein is the most highly conserved NS protein among flaviviruses, and thus a preferred target for development of antiviral drugs against DENV and flaviviruses in general [59]. Though its role in WNV virulence in mammals has been documented [31], no data are available for NS5 in birds. For all these reasons, we decided to compare parental WNV-IS98 and IT08 strains and two chimeric constructs thereof, IS98-NS4A/NS4B/5′NS5 and IS98-NS5/3′UTR, in vitro and in vivo and in both mammalian and avian models, to elucidate host-dependent virulence factors.

Measurement of growth kinetics in the duck embryo cell line CCL141 revealed that early replication of IS98-NS4A/NS4B/5′NS5 was more efficient than that of WNV-IS98. Indeed, higher viral RNA loads were obtained at 17 hpi with this chimera, suggesting enhanced molecular mechanisms during the first steps of virus replication in avian cells. However, no differences in viral titer were observed for these viruses. Increase in total viral RNA quantity without a corresponding increase in infectious viral titers in cell supernatants could be explained by increased production of defective viral particles, or the release of unpackaged viral RNA into the supernatant. Synonymous mutations identified in chimeras could also modify interaction of viral RNA with viral or cellular proteins, possibly altering RNA quantity without changing viral infectious titer. In SPF chickens, this chimeric virus proved to be less virulent than WNV-IS98, displaying a lower viral load as quantified in oral swabs at 7 dpi (*p* < 0.05) and a lower mortality rate at 14 dpi (29% for IS98-NS4A/NS4B/5′NS5 vs. 100% for WNV-IS98). Of note, the IS98-NS4A/NS4B/5′NS5 chimera and the parental WNV-IT08 strain induced a similar mortality rate. Our in vitro and in vivo results thus reveal similarities in the replicative features of IS98-NS4A/NS4B/5′NS5 and WNV-IT08 strains in avian models, suggesting that the NS4A/NS4B/5′NS5 region could play a role in the attenuated phenotype of WNV-IT08 compared with WNV-IS98 in birds.

We also conducted experiments in mammalian models, including in vitro assays on Vero cells and virulent challenge in BALB/cByJ mice, to evaluate host-dependent responses to genetically modified viruses. In vitro analysis of growth kinetics argued in favor of an attenuated phenotype for the IS98-NS5/3′UTR chimera, as shown by smaller plaque sizes and lower viral RNA production relative to those of parental viruses and the IS98-NS4A/NS4B/5′NS5 chimera (Figure 3 and Figure 4). In vivo, peak viremia in IS98-NS5/3′UTR-infected mice at 3 dpi was significantly lower than that elicited by other viruses. Moreover, mortality rates at 21 dpi in BALB/cByJ mice suggested that WNV-IS98 was more virulent than the IS98-NS5/3′UTR chimera (100% and 40%, respectively), although the difference did not attain statistical significance.

We sought to predict which substitutions in the IS98-NSA4/NS4B/5′NS5 chimera were likely to be responsible for its attenuated phenotype in avian models, especially in SPF chicks. We performed a literature review to this effect, whose results are presented in Table 1. In reference to previous studies, the replacement of the glutamic acid residue at position 249 by glycine (NS4B-E249G) would appear to be a good candidate. Indeed, this substitution has not only been identified in an attenuated strain isolated from birds in 2003 in Texas [60], but was also shown to be crucial for conferring an attenuated phenotype in mice [61]. A second amino acid residue in this region, NS4A-92, is known to be implicated in avian virulence (Table 1), but since it is conserved between WNV-IS98 and IT08 strains, it cannot play a role in their differential virulence.

Regarding virulence in mammals, the attenuated phenotype of IS98-NS5/3′UTR observed in mice could be due to mutations in the NS5 protein, whether synonymous or non-synonymous, or in the 3′UTR region. The NS5 protein, which is the largest of the NS proteins, is responsible for the capping and replication of viral RNA, catalyzed by its N-terminal methyltransferase (MTase) and C-ter RNA-dependent-RNA-polymerase (RdRp), respectively [62]. WNV-IS98 and IT08 differ by six non-synonymous mutations in this region: NS5-H53Y, NS5-S54P, NS5-V258A, NS5-N280K, NS5-A732V, and NS5-R422K. Although none of these have been implicated as yet in WNV virulence in mammals, our results suggest that they would be worth investigating as candidates for the attenuated phenotype of IS98-NS5/3′UTR observed in mice. Moreover, the synonymous mutations in NS5 could also have an impact on virulence by affecting the stability or structure of viral RNA, which could potentially modify interactions between viral RNA and viral or cellular proteins and exert an impact on viral replication.

As NS5 is highly conserved among flaviviruses, we gave this region particular attention for investigation of the differential virulence of WNV-IS98 and IT08, focusing on genetic markers that have been described to determine pathogenicity of other flaviviruses, such as DENV. Several NS5 markers are known to be responsible for enhanced replication and virulence of flaviviruses, mainly DENV, including (i) K-D-K-E catalytic tetrad (NS5-K61-D146-K182-E218), (ii) single mutations NS5-R325A, NS5-R519A, NS5-R769A, NS5-K840A, NS5-R841A, (iii) PDZ-Binding Motifs (PBM motif), and (iv) post-translational modifications. (i) The K-D-K-E tetrad catalyzes 2′O methylation and is highly conserved among flaviviruses. Disruption of the tetrad confers an attenuated phenotype, providing a promising strategy for development of a live attenuated DENV vaccine [57,63]. (ii) Iglesias et al. [64] showed that the listed single mutations NS5-R325A, NS5-R519A, NS5-R769A, NS5-K840A, and NS5-R841A impart high RNA synthesis activity in vitro but delayed or impaired replication in vivo for DENV. (iii) The PBM motif at the N-terminus of the NS5 protein is widely known to be implicated in virulence. In WNV it comprises three amino acids, T-V-L, just upstream of the stop codon. The PBM motif is known to bind cellular proteins possessing a globular domain known as the PDZ domain. As interactions between NS5 and host proteins have been poorly characterized, Giraud et al. [65] investigated interaction between the NS5 PBM domain of WNV and PDZ-proteins, both in vitro and in vivo, and demonstrated that the C-terminal PBM of WNV NS5 recognizes several human PDZ. (iv) Several post-translational modifications of flavivirus NS5—including threonine phosphorylation—could have potential regulatory roles. NS5 phosphorylation is broadly conserved in flaviviruses [66,67]. The DENV NS5 protein is phosphorylated by both mammalian and mosquito protein kinase G at the conserved Thr449 residue. This phosphorylation is essential for successful virus propagation. To analyze the potential role of NS5 markers in the attenuated chimera IS98-NS5/3′UTR, we aligned the sequences of the two parental WNV strains, IS98 and IT08, with that of a representative DENV-2 strain; the virulence of DENV-2 being extensively characterized (Appendix A Figure A1). No differences were noted among the two parental strains and DENV-2 at these positions, suggesting that the attenuated phenotype of IS98-NS5/3′UTR could not be explained by the known molecular markers of virulence in NS5. The NS5 protein also plays a key role in evasion of the host immune response, notably by inhibiting the JAK-STAT pathway and consequently induction of the type I IFN response [68]. Synonymous mutations in this region could therefore affect the mammalian immune response, as has been documented for several flaviviruses including ZIKV, JEV, and DENV. ZIKV NS5 inhibits the phosphorylation of STAT1 and leads to degradation of STAT2, while JEV, DENV, or WNV have been shown to antagonize this pathway too [68]. STAT2 is frequently targeted for NS5-mediated IFN suppression in flaviviruses, and modification in NS5 nucleotide sequence could consequently impact host immune response and lead to more efficient viral infection [69]. IS98-NS5/3′UTR and WNV-IT08 caused reduced mortality in mice at 21 dpi compared with WNV-IS98. It is possible that WNV-IT08 and IS98-NS5/3′UTR present molecular features that failed to inhibit host innate immune responses, especially type-I IFN response, and in this context, the role of non-synonymous NS5-V258A, NS5-N280K, NS5-A372V, and NS5-R422K mutations could be investigated.

Furthermore, as the IS98-NS5/3′UTR chimera possesses not only the NS5 coding region but also the 3′UTR of WNV-IT08, the attenuated phenotype in mice could be related to modifications in the 3′UTR region. The flavivirus genome is flanked by the 5′UTR and 3′UTR. The 5′UTR is about 100 nt long, whereas the 3′UTR differs in length between flaviviruses, ranging from 400 to 700 nt. Interaction between 5′UTR and 3′UTR is critical for viral RNA replication [70], notably via its recruitment of the NS5 polymerase. Only a single study has addressed the role of the 3′UTR region in WNV host virulence [48]. As the 3′UTR plays a role in viral replication, it would be interesting to address its possible role in the differences observed between parental viruses and chimeras. Moreover, it is possible that the 5′UTR and 3′UTR of the two strains, WNV-IS98 and IT08, are incompatible, which could impair their interaction/cyclization and ultimately perturb viral replication for the IS98-NS5/3′UTR chimera. Sequence alignment of WNV-IS98 and IT08 strains revealed 25 mutations in their 3′UTR. The 3′UTR is highly structured with regions conserved between flaviviruses. It is composed of a stem-loop (SL), two dumbbell (DB) structures, conserved sequences (CS), and repeated conserved sequences (RCS). One of the important nucleotide sequences in the 3′UTR is the 3′-terminal stem-loop (3′SL). This region of the RNA genome of flavivirus includes sequence elements that are required for genome cyclization. Cyclization is a prerequisite for the initiation of viral replication. One part of 3′SL is metastable and confers a structural flexibility that could be implicated in change from linear to circular RNA. Many studies show a role of 3′SL in RNA replication [71]. None of the 25 3′UTR mutations found between WNV-IS98 and IT08 are in 3′SL, suggesting that the difference of viral replication between WNV-IS98 and IT08 could not be explained by a difference in cyclization depending on the 3′SL sequence (Table 1). The 3′UTR sequence has been more extensively investigated for DENV. Manzano et al. created deletions of five nucleotides in the DB sequence [72], showing that viral translation rate was reduced by 60% with a double deletion mutant nt 10,474–10,478/nt 10,562–10,566 in mammalian cells. Sequence alignment of the 3′UTR region of the DENV-2 virus and parental strains, WNV-IS98 and IT08, showed 100% identity in the 10562–10566 sequence between the four viruses (Appendix A Figure A3). However, the 10,474–10,478 region differs by one nucleotide between WNV-IS98 and IT08. Its role in viral translation for Dengue virus could suggest that this region could similarly influence viral translation of WNV-IS98 and IT08 and the corresponding chimeras (Table 1).

We chose to create viral chimeras between two Mediterranean strains, WNV-IS98 and IT08, to identify which molecular regions could be implicated in differences in bird mortality during WNV outbreaks. In an SPF chick model, IS98-NS4A/NS4B/5′NS5 and WNV-IT08 displayed a similarly attenuated behavior, whereas IS98-NS5/3′UTR seemed to be as virulent as WNV-IS98. Our results suggest that the NS4A/NS4B/5′NS5 region, and probably the NS4B-249 residue, could be implicated in decreased virulence of WNV in SPF chicks and possibly in slightly decreased virulence in mice. The NS5-3′UTR region seemed to decrease virulence in mammalian hosts, possibly in relation to modulation of the NS5 polymerase activity or of host innate immunity. Taken together, our results suggest that genome modifications have different effects depending on the nature of the host [73] and provide new molecular determinants candidates in WNV virulence. It is already known that molecular markers of virulence can be host-dependent, as previously illustrated by Puig-Basagotti et al., who showed that the NS4B-E249D mutation modified virulence in mammalian but not in mosquito cell models. This analysis, however, was performed only in vitro. Here we sought to improve understanding of the role of molecular markers in different in vivo hosts. Modifications in the 3′ end genomic region could influence protein–protein interactions (PPI) and notably between NS4A/NS4B and NS2A and NS2B, or between NS3 and NS5 proteins [74]. These interactions have also been described for DENV [75] between NS2B/NS3 [76], NS4A/NS4B [77], NS1/NS4B [78], NS3/NS4B [79], and NS3/NS5 [80]. Finally, not only protein–protein, but also viral RNA-protein interactions, could underlie the modification in virulence.

Although our study focused on the 3′ end of the WNV genome, it would be interesting to investigate other NS and structural proteins of these two Mediterranean strains, and notably the E-159 residue known to be a virulence marker and presenting a non-synonymous mutation between WNV-IS98 and IT08 [81]. Moreover, only a few studies have addressed WNV virulence in mosquitoes, and WNV infections in mosquitoes would substantially improve our understanding of viral transmission dynamics. Finally, our study provides new insight into genetic markers that may influence the virulence of these two Mediterranean strains in birds and in mammalian hosts.

**Table 1 viruses-15-01094-t001:** Comparison of known literature data with our results. Possible role of known molecular flaviviruses virulence markers in modifying chimeras IS98-NS4A/NS4B/5′NS5 and IS98-NS5/3′UTR virulence in avian or mice models.

Genome Sequence	Literature Known Molecular Markers	Role	References	Comparison to Our Study	Conclusion for Our Study
NS4	NS4A-F92L	Increase virulence in birds	[58]	NS4A-92 amino acid is the same between IS98 and IT08 so could not explain any difference in chicks	No role of NS4A-92 amino acid in modulating virulence in chicks
NS4	NS4B-E249G	The substitution of the glutamic acid at position 249 (E249) was shown to be crucial for conferring an attenuated phenotype in mice and it was identified in an attenuated strain isolated from birds in 2003 in Texas	[60,61]	RNA quantity in oral swab if IT08 and IS98-NS4A/NS4B/5′NS5 are significantly lower than IS98 and IS98-NS5/3′UTR at 7 days p.i. → suggesting a role of NS4A/NS4B/5′NS5 in reduced virulence in birds IT08 possesses NS4B-D249 and IS98 possesses E249	Possible role of NS4B-E249D mutation in IT08 or chimera IS98-NS4A/NS4B/5′NS5 in attenuated virulence in chicks
NS5	K-D-K-E motif (NS5-K61-D146-K182-E218)	Candidate for DENV vaccine	[57,63]	No difference in K-D-K-E motif between IS98 and IT08 (Appendix A Figure A1)	Known molecular markers of virulence of DENV-2, a flavivirus closely related to WNV, cannot, by extension, explain differences observed between WNV IS98 and IT08 and corresponding chimeras
NS5	NS5-R325A, NS5-R519A, NS5-R769A, NS5-K840A, NS5-R841A	High RNA synthesis activity in vitro but delayed or impaired replication in vivo	[64]	Sequence alignment between DENV-2 and our parental strain does not show any difference between IS98 and IT08 (Appendix A Figure A1) 100% identity between IS98/IT08 at these sites and DENV-2 virus, confirming their potential role in flavivirus replication	Known molecular markers of virulence of DENV-2, a flavivirus closely related to WNV, cannot, by extension, explain differences observed between WNV IS98 and IT08 and corresponding chimeras
NS5	PBM motif (three amino acids, T-V-L), just upstream the codon stops	C-term PBM of WNV NS5 recognizes several human PDZ PBM is known to be implicated in virulence	[65]	Sequence alignment of IS98 and IT08 does not show any difference in PBM motif (Appendix A Figure A1)	Known molecular markers of virulence of DENV-2, a flavivirus closely related to WNV, cannot, by extension, explain differences observed between WNV IS98 and IT08 and corresponding chimeras
NS5	Threonine phosphorylation (Thr449)	Regulatory roles	[66,67]	Sequence alignment of IS98 and IT08 does not show any differences in PBM motif (Appendix A Figure A2)	Known molecular markers of virulence of DENV-2, a flavivirus closely related to WNV, cannot, by extension, explain differences observed between WNV IS98 and IT08 and corresponding chimeras
NS5	General modifications in NS5	NS5 protein is known to frequently use multiple strategies to suppress the JAK-STAT signaling pathway and consequently type I IFN response to mammalian NS5. ZIKV inhibits the phosphorylation of STAT1 and leads to degradation of STAT2; JEV, DENV, and WNV antagonize this pathway too STAT2 is frequently targeted for NS5-mediated IFN suppression in flaviviruses, so modifications in NS5 could impact host immune response to viral infection	[69]	IS98-NS5/3′UTR and IT08 engender reduced mortality in mice at 12 days p.i. compared to IS98 It is possible that IT08 and IS98-NS5/3′UTR present molecular features that could not inhibit host innate response	Possible role of NS5 in decreased virulence of IT08 in a mammalian model, maybe by influencing host IFN response
3′UTR	3′-terminal stem-loop (3′SL)	Interaction between 5′UTR and 3′UTR is implied in viral RNA replication. This interaction is important for NS5 polymerase recruitment. Role of 3′SL in viral replication of DENV	[71]	None of the 25 3′UTR mutations found between IS98 and IT08 are in 3′SL, suggesting that the difference of viral replication between IS98 and IT08 could not be explained by a difference in cyclization depending on 3′SL sequence	No role of 3′SL sequence in attenuation of virulence of IT08 or IS98-NS5/3′UTR in a mammalian model
3′UTR	Double deletion mutant nt 10,474–10,478/nt 10562–10566 in DENV	Viral translation rate was reduced by 60% in mammalian cells	[72]	Sequence alignment of 3′UTR region of DENV-2 virus and parental strain, IS98 and IT08 show 100% identity in the 10562–10566 sequence between the four viruses (Appendix A Figure A3) However, 10,474–10,478 differs in one nucleotide between IS98 and IT08. Nucleotide 10,478 of IT08 is the same as DENV-2 (nt T), whereas it is different in IS98 (nt C). That could impact viral translation	Possible role of nucleotide 10,478 in reduced virulence of IT08 and IS98-NS5/3′UTR in a mammalian model

## Figures and Tables

**Figure 1 viruses-15-01094-f001:**
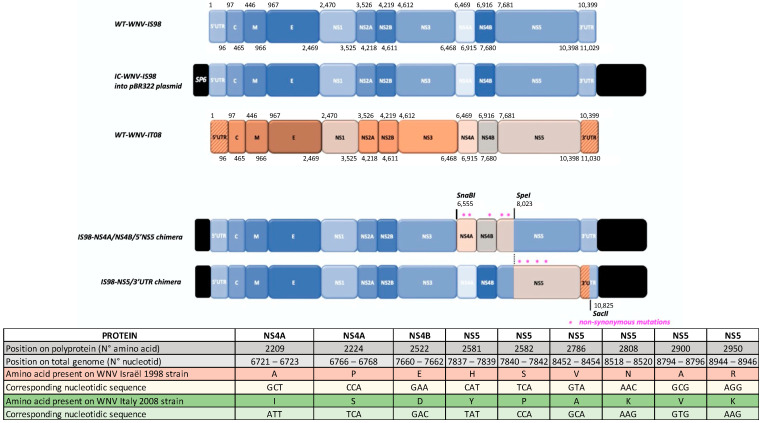
Genomic positions of genomic fragments exchanged between WNV IS98 and IT08 strains in IS98-NS4A/NS4B/5′NS5 and IS98-NS5/3′UTR chimeras. Blue and orange fragments originate from IS98 and IT08 genomes, respectively. Non-synonymous mutations are represented by pink stars. Restriction enzyme sites used for cloning (SnaBI, SpeI, SacII) are indicated.

**Figure 2 viruses-15-01094-f002:**
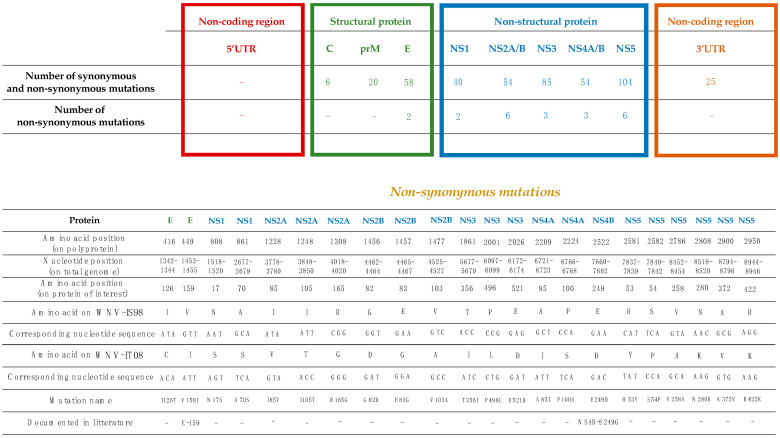
Number and location of synonymous and non-synonymous mutations between WNV-IS98 and WNV-IT08.

**Figure 3 viruses-15-01094-f003:**
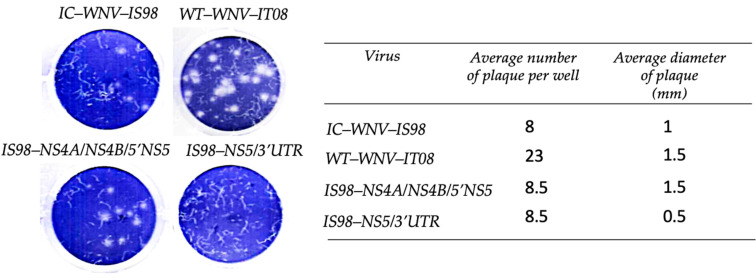
Comparison of plaque size and average number of plaques for IC-WNV-IS98 and WT-WNV-IT08 parental strains and chimeras in Vero cells, at 3 dpi.

**Figure 4 viruses-15-01094-f004:**
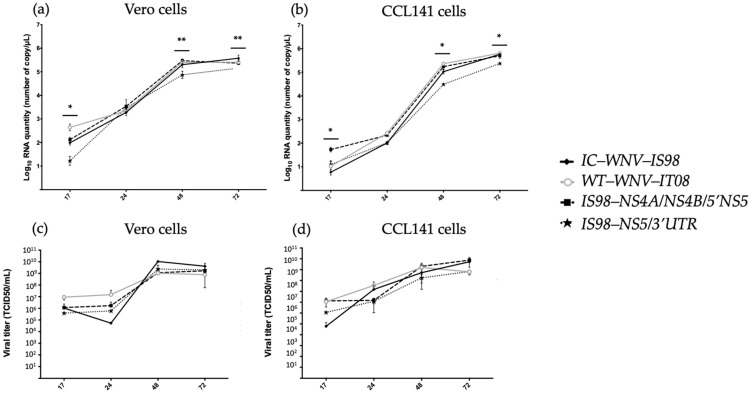
Growth kinetics of IC-WNV-IS98 and WT-WNV-IT08 parental strains and IS98-NS4A/NS4B/5′NS5 and IS98-NS5/3′UTR chimeras thereof. Vero and CCL141 cells were infected with the indicated viruses at a multiplicity of infection (MOI) of 0.1. At the indicated time points, cell culture supernatants were collected and their viral RNA content was analyzed either by RT-qPCR (**a**,**b**) or by end-point dilution assay (**c**,**d**). Statistical comparisons were performed by the non-parametric Kruskal–Wallis test. The error bars indicate standard deviations (SD). *, *p* < 0.05; **, *p* < 0.01.

**Figure 5 viruses-15-01094-f005:**
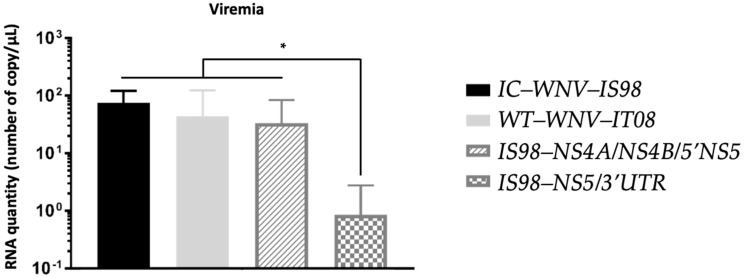
Viremia 3 days after intraperitoneal inoculation of female BALB/cByJ mice with 1 PFU of IC-WNV-IS98, WT-WNV-IT08, and chimeras. Viral load in blood was quantified by RT-qPCR. *, *p* < 0.05.

**Figure 6 viruses-15-01094-f006:**
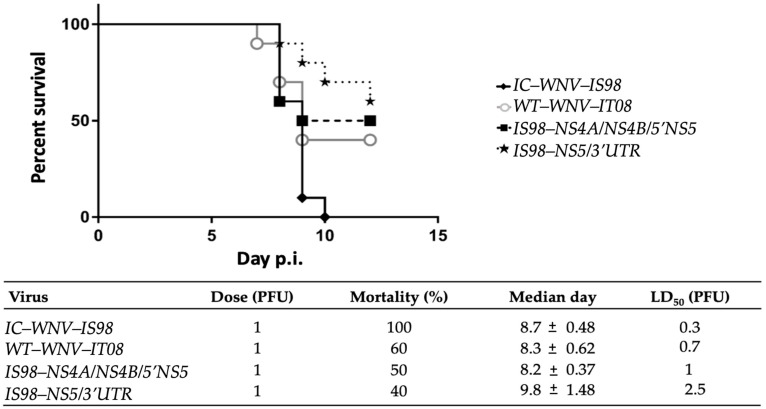
Neurovirulence of IC-WNV-IS98, WT-WNV-IT08, and chimeras. Survival of 6-week-old female BALB/cByJ mice after i.p. inoculation with 1 PFU of parental strain or chimeras is shown. LD_50_ determination after i.p. inoculation of BALB/cByJ mice with varying virus doses is also presented.

**Figure 7 viruses-15-01094-f007:**
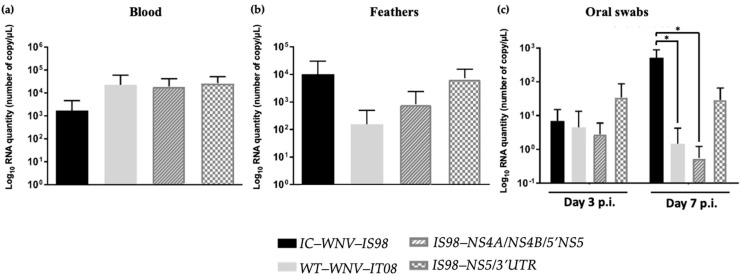
Viral titers in (**a**) blood at 3 dpi, (**b**) feathers at 3 dpi, (**c**) oral swabs of one-day-old SPF chickens at 3 and 7 dpi after i.c. inoculation of 10^3^ TCID_50_ of parental strains and chimeras. *, *p* < 0.05.

**Figure 8 viruses-15-01094-f008:**
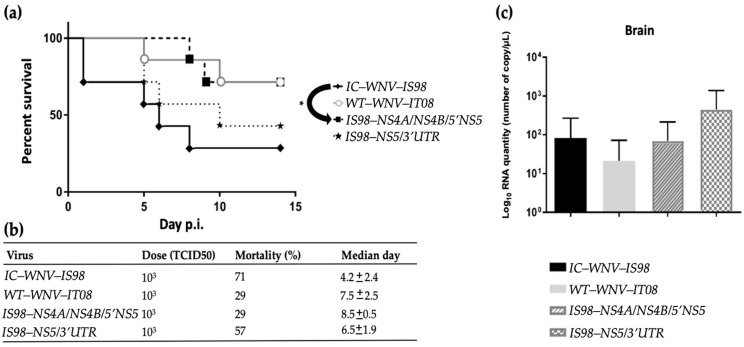
Neurovirulence of IC-WNV-IS98, WT-WNV-IT08, and their chimeras in one-day-old SPF chickens. Chickens were infected intracerebrally with 10^3^ TCID_50_ and monitored for 14 dpi. (**a**) Survival growth. (**b**) Mortality and median day of death are presented. (**c**) Viral titer in the brain of dead chicks assessed by RT-qPCR. *, *p* < 0.05.

## Data Availability

Data obtained during this study can be obtained by contacting the authors.

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
