# Peer review of "Evaluation of NS4A, NS4B, NS5 and 3′UTR Genetic Determinants of WNV Lineage 1 Virulence in Birds and Mammals"

_viruses, 2023, doi:10.3390/v15051094_

Round 1
Reviewer 1 Report
- Materials and methods section 2.2 Generation of chimeric viruses is missing a reference to Figure 1.
- Materials and methods section 2.5 Provide the names of the target genes for the PCR, and the length of the target fragments.
- Materials and Methods section 2.7 Provide more details about the animal experiment in mice (the number of groups, the number of animals in the group, collection of samples, scheduled necropsies).
- Materials and methods section 2.8 The same as in 2.7 plus provide details of the collection and processing of fevers.
- Results, section 3.2. A comment: The first sentence suggests that small plaques correlate with reduced virulence of the virus. This contradicts the provided data, which indicates that the virulent strain IS98 has a smaller plaque size compared to the low virulent strain IT08.
Author Response
We thank both reviewers for their valuable comments, that allowed us to improve our manuscript. You will find below a list of modifications brought to the manuscript to be considered by the reviewers.
Reviewer 1:
- Materials and methods section 2.2 Generation of chimeric viruses is missing a reference to Figure 1.
Figure 1 is cited in the materials and methods section 2.2 on line 160. Figure 1 was prepared for the manuscript by one of the first author, Lise Fiacre. No modification was consequently brought to the text.
- Materials and methods section 2.5 Provide the names of the target genes for the PCR, and the length of the target fragments.
The genomic region targeted by the WNV PCR and the length of WNV and ACTB amplicons were added in the manuscript. The following information was added on line 206 for the WNV RT-PCR “targeting the 5’UTR and C regions (total length : 123 nucleotides)” and on lines 210-211 for the ACTB RT-PCR “(total length of the ACTB amplicon : 131 nucleotides)”.
- Materials and Methods section 2.7 Provide more details about the animal experiment in mice (the number of groups, the number of animals in the group, collection of samples, scheduled necropsies).
The corresponding details on WNV experimental infections in Balb/cByJ mice were provided in section 2.7, lines 232-234. Blood and brain tissues were collected at 3dpi and after animal death respectively, as mentioned in the manuscript. RNA extraction methods have been detailed lines 239-241.
- Materials and methods section 2.8 The same as in 2.7 plus provide details of the collection and processing of fevers.
The corresponding details on WNV experimental infections in SPF chicks were provided in section 2.8, lines 248-265.
- Results, section 3.2. A comment: The first sentence suggests that small plaques correlate with reduced virulence of the virus. This contradicts the provided data, which indicates that the virulent strain IS98 has a smaller plaque size compared to the low virulent strain IT08.
Small plaques are usually associated with reduced virulence and viral growth; however, such observations are not systematically reported, depending on the cell line used, the experimental design (incubation temperature or length…). WNV-IT08 tends to develop larger plaques (1,5mm) than WNV-IS98 (1mm), although this finding is not statistically significant. Similarly, we report that IS98-NS5/3’UTR tends to produce smaller plaques than other parental or chimeric viruses, that would be indicative of an attenuated phenotype.
The paragraph (lines 298-309) was modified to respond the reviewer’s comment, and specifically the sentence “Noteworthy, WNV-IT08 demonstrating lower virulence than WNV-IS98 in an avian model of WNV infection (SPF chicks) did not show reduced virulence in Vero cells.” was added.

Reviewer 2 Report
The article "Evaluation of NS4A, NA4B, NS5 and 3'UTR genetic determinants of WNV lineage 1 virulence in birds and mammals" investigates the role of non-structural proteins on the properties of two WNV isolates. This is achieved by introducing the NS proteins (in two blocks NS4A & B or NS5/3'UTR) from an Italian isolate IT08 into a backbone from an isolate from Israel IS98. The authors argue that IS98 represents a highly virulent strain whereas IT08 is less so although it would be helpful to identify the host from which both were isolated from. Whilst the aims and execution of the article are good, there are a number of points that need explanation. These are:
1) The reference to Bahuon et al is given as superscript 47 (not journal style) and is [49] in References. No reference given for Reed and Meunch on page 5. On page 6, Dridi et al is given in superscript as 34 but is not in the References. The authors should revise the references to correct this. Also CO2 on pg5.
2) The authors state on pg7 that "The efficacy of viral cell-to-cell spread, and hence the size of viral plaques, is correlated with virulence." Can the authors provide a reference for this and explain how this can be true in light of data in Figure 6. Also no statistical analysis of plaque sizes data in Figure 3. The growth curves in Figure 4 may suggest minor variations in growth kinetics but visually they look very similar.
3) For animal experiments, can the authors give group sizes throughout. Can they also explain why data is only given to d12 in Figure 6, whilst the M&Ms suggest that experiments went to Day 21. They also suggest serology was conducted but this is not presented in the article.
4) In Figure 8, the group receiving IS98 wild type appears to lose 30% at day 1. This seems unlikely to be due to the infection process and without this drop, all the survival curves would be similar. Can the authors comment? As above, the M&M suggest a subcutaneous inoculation in SPF chickens but this data is not presented.
5) The discussion is far to long, especially with the addition of an extensive table comparing previous studies with this data set. Suggest revision to reduce this.
Overall the study suggests that NS proteins play a role in WNV virulence. The data presented supports this although does not provide a mechanism.
Author Response
We thank both reviewers for their valuable comments, that allowed us to improve our manuscript. You will find below a list of modifications brought to the manuscript to be considered by the reviewers.
Reviewer 2:
-The authors argue that IS98 represents a highly virulent strain whereas IT08 is less so although it would be helpful to identify the host from which both were isolated from.
The corresponding information was added in the materials and method section, lines 144-151.
-The reference to Bahuon et al is given as superscript 47 (not journal style) and is [49] in References. No reference given for Reed and Meunch on page 5. On page 6, Dridi et al is given in superscript as 34 but is not in the References. The authors should revise the references to correct this. Also CO2 on pg5.
We corrected the citations in the manuscript, lines 148 and 291 and added reference 52 relative to Reed and Muench methodology. CO2 was corrected on line 196.
-The authors state on pg7 that "The efficacy of viral cell-to-cell spread, and hence the size of viral plaques, is correlated with virulence." Can the authors provide a reference for this and explain how this can be true in light of data in Figure 6. Also no statistical analysis of plaque sizes data in Figure 3. The growth curves in Figure 4 may suggest minor variations in growth kinetics but visually they look very similar.
Citation 53 was added in the manuscript and the paragraph was modified to consider the reviewer’s comment. Small plaques are usually associated with reduced virulence and viral growth; however, such observations are not systematically reported, depending on the cell line or model used, the experimental design (incubation temperature or length in cell culture assays for example). The sentence “Noteworthy, WNV-IT08 demonstrating lower virulence than WNV-IS98 in an avian model of WNV infection (SPF chicks) did not show reduced virulence in Vero cells.” was added
lines 308-309. Growth curves in Figure 4 show similar patterns but present statistical differences, demonstrating differences in viral replication between parental and chimeric viruses (IS98-NS5/3’UTR chimera specifically) in Vero and CCL41 cell lines.
- For animal experiments, can the authors give group sizes throughout. Can they also explain why data is only given to d12 in Figure 6, whilst the M&Ms suggest that experiments went to Day 21. They also suggest serology was conducted but this is not presented in the article.
Group sizes were added on lines 232-234 (mice) and 248-265 (birds). Mice were monitored for the presence of clinical signs for 14 days, as represented on figure 6 – such a clinical follow-up allows to observe WNV clinical disease in infected mice, as previously shown in [49]. In order to demonstrate WNV infection in surviving mice, WNV ELISA was performed on mice sera sampled on day 21 post-infection; the corresponding data are not shown in the manuscript, but we added a sentence in the materials and method section to present the major conclusion driven from these assays (lines 245-246).
-In Figure 8, the group receiving IS98 wild type appears to lose 30% at day 1. This seems unlikely to be due to the infection process and without this drop, all the survival curves would be similar. Can the authors comment? As above, the M&M suggest a subcutaneous inoculation in SPF chickens but this data is not presented.
Intracerebral inoculation of WNV in SPF chicks is associated with rapid onset of neurological clinical signs and rapid death (in less than 24h after disease onset). In earlier assays describing WNV infection with several WNV strains by the IC route on SPF chicks, death was reported as soon as 2 days post-infection [36]. Earlier death at 1dpi could be indicative of a bad tolerance of the IC injection in the animals, but in this latter case, we would have observed early deaths in the other groups as well, while no death were reported in the PBS control group and mortalities were reported from 5dpi up to 10 dpi in the groups infected with the 3 other viruses; we can assume that early death was specifically induced in the WNV-IS98 group in our assays. Moreover, our conclusions are drawn from mortality rates at the end of the experiment and even when focusing on mortalities observed after day 5 post-infection, the same conclusions were obtained, with WNV-IS98 and IS98-NS5/3’UTR inducing higher mortality rates that WNV-IT08 and IS98-NS4A/NS4B/5’NS5.
The materials and method section was modified and mention of the subcutaneous inoculation of SPF chickens was removed on lines 252-253.
_The discussion is far to long, especially with the addition of an extensive table comparing previous studies with this data set. Suggest revision to reduce this.
The study of molecular interactions or mechanisms involved the modulation of WNV virulence would deserve additional studies. Because no mechanistic analyses were performed in this study, we took the opportunity of the discussion section to propose hypothesis and understand the differences observed between parental and chimeric viruses. Molecular markers of virulence established in the literature were considered and their contribution thoroughly analyzed (Table 1). We think that table 1 is helpful to guide the discussion and that the whole discussion is important to discuss the results obtained and drive further mechanistic analysis. We did not bring modifications to the discussion.
